# Impacts of Human Activity and Climate Change on the Suitable Habitats for *Xanthium spinosum* in China

**DOI:** 10.3390/plants14030306

**Published:** 2025-01-21

**Authors:** Yabin Liu, Yuyu Li, Rui Wang, Lizhu Guo, Yu Ji, Yihao Chen, Lifen Hao, Kejian Lin

**Affiliations:** 1Key Laboratory of Agri-Products Quality and Biosafety, Ministry of Education, Anhui Provincial Key Laboratory of Integrated Pest Management on Crops, School of Plant Protection, Anhui Agricultural University, Hefei 230036, China; jt5240035@gmail.com; 2Key Laboratory of Biohazard Monitoring and Green Prevention and Control for Artificial Grassland, Ministry of Agriculture and Rural Affairs, Grassland Research Institute, Chinese Academy of Agricultural Science, Hohhot 010019, China; liyuyu@caas.cn (Y.L.); ellenguo@sina.cn (L.G.); jiyu1998imu@163.com (Y.J.); 18147012964@163.com (Y.C.); 3State Key Laboratory for Biology of Plant Diseases and Insect Pests, Institute of Plant Protection, Chinese Academy of Agricultural Sciences, Beijing 100193, China; wangrcaas@163.com

**Keywords:** biomod2, invasive alien plants, species distribution model, future climate, *Xanthium spinosum*

## Abstract

*Xanthium spinosum* (*X. spinosum*) is a highly invasive weed native to South America and distributed in 17 provinces (municipalities) of China. It has severely negative influences on ecosystems, agriculture, and husbandry. However, few studies have reported on the impact of human activity and climate change on the future distribution and centroid shift of *X. spinosum*. This study aimed to investigate the potential geological distribution of *X. spinosum* in China, as well as the distribution pattern, centroid shift, and key environmental factors influencing its distribution, under four future climate scenarios (SSP1-26, SSP2-45, SSP3-70, and SSP5-85) based on the biomod2-integrated model. The results indicated that the suitable habitats for *X. spinosum* would expand in the future, mainly in Inner Mongolia, Northeast China, and the plateau regions (e.g., Xinjiang and Xizang). Under future climate scenarios, the centroid would shift toward the northwest or northeast part of China, with the SSP2-45-2050s scenario showing the maximum shift distance (161.990 km). Additionally, the key environmental variables influencing the distribution of *X. spinosum*, including human impact index, bio5, bio7, and bio12, were determined, revealing that most of them were related to human activities, temperature, and precipitation. This study enhances the understanding of the influence of human activity and climate change on the geographic range of *X. spinosum*. It provides references for early warning and management in the control of *X. spinosum*.

## 1. Introduction

Biological invasion refers to the process by which species that originally do not belong to a certain ecological or geographic region are spread to this new region through different channels and colonize, establish, expand and spread in the new habitat, bringing certain negative impacts on the economy and ecology of the place of introduction [1]. It has attracted attention globally because it threatens biodiversity, ecosystem functions, and services and significantly impacts the environment and economy of invaded regions [2]. When an alien species arrives in a new region, it is subjected to heterogeneous environmental conditions. Its population grows and expands rapidly in an optimized habitat but slowly under adverse climatic conditions; however, it can rapidly expand when the climatic conditions become favorable. Therefore, the successful invasion of alien species is closely related to the local climate, environment, and human activities. The global average surface temperature has increased over the past 100 years by approximately 0.6 °C [3]. Climate change, by raising temperatures, altering precipitation patterns, and increasing the frequency of extreme weather events, has forced invasive species to undergo adaptive evolution. In addition to climate change, human activities such as global trade, transportation, and agricultural practices have introduced many species into non-native regions. For instance, bamboo, as an economically important plant, has been widely used for construction, food, and ornamental purposes, resulting in its introduction to numerous new environments. However, these introduced species can sometimes become invasive in their new habitats, disrupting native ecosystems [4]. Moreover, the interaction between human-induced habitat disturbances and climate change has amplified the ecological impacts of biological invasions [5]. Thus, predicting the potential distribution of invasive alien species based on information on human activities and future climate change is critical for the early monitoring and warning of biological invasions.

*Xanthium spinosum* L. [6] is native to the subtropical regions of South America [7] and later appeared as wild naturalized species in central and southern Europe, Asia, and North America [8]. In 1932, it was first discovered in Dancheng County, Henan Province [9]. Recent studies have demonstrated that *X. spinosum* is distributed in 17 provinces in China, including Beijing, Inner Mongolia, Hebei, Henan, and Anhui [10,11,12]. It has a long flowering phase, a strong reproductive capacity with high seed production, and considerable cold tolerance [13,14]. Its involucres have hooked spines, facilitating its spread through vectors such as cattle, sheep, and goods and reducing the value of livestock hairs and hides [15]. *X. spinosum* can exert allelopathic effects on native plants [16,17] and negatively impacts various crops such as beet (*Beta vulgaris* L.), soybean (*Glycine max* (L.) Merr.), cotton (*Gossypium* L.), sunflower (*Helianthus annuus* L.), wheat (*Triticum aestivum* L.), and maize (*Zea mays* L.) [18]. Once it successfully invades and establishes populations, it can severely affect the species diversity, agriculture, and animal husbandry of the invaded region, while significantly increasing the costs of ecological restoration and economic management. In the context of intensified climate change, identifying the potential distribution of invasive species and their responses to environmental changes is crucial. This can not only enhance early detection and early warning capabilities for invasion risks but also provide a scientific basis for assessing agricultural economic losses and formulating management strategies for invasions.

Previous studies on *X. spinosum* focused on its bio-ecological characteristics [14,19], invasion mechanisms, and diffusion modes [17,20,21]. However, these studies rarely predicted the distribution of suitable habitats for *X. spinosum* under climate change based on the species distribution model (SDM). Thus, clarifying the potential geological distribution of *X. spinosum* and its response to climate change is of great significance for its monitoring, early warning, and control.

The SDM can be used to associate species distribution with environmental information, enabling the analysis of suitable distribution ranges and potential invasion risk regions [22,23]. Since its inception, SDM has been widely used to predict species distribution across terrestrial, riverine, and marine domains [22,24,25,26]. Commonly used single models in this field include the regression-based generalized linear model (GLM) [27], the machine learning-based random forest (RF) model [28], and the maximum entropy [29] (MAXENT) model are commonly used. The selection of different models with varying strengths and weaknesses may lead to variations in prediction results for the same species data, negatively affecting policy decision-making. Therefore, an “ensemble forecasting” was proposed, wherein multiple models were constructed using different modeling methods and then integrated for prediction [30,31]. Wu et al. [32] conducted a review and synthesis of ensemble forecasting methods, revealing that ensemble forecasting generally enhances prediction accuracy and stability across various fields (e.g., weather, economics, and political forecasting). This improvement is especially evident in its origins in weather forecasting. Further, through applications of ensemble forecasting in transportation-related cases, they highlighted the potential of these models to improve both prediction accuracy and reliability.

Biomod2, developed in the R language environment, is a predictive platform that integrates multiple species distribution models [33,34]. Recently, suitable habitats for invasive alien plants based on an integrated model have gained the attention of researchers. Recent studies on the potential geological distribution of more than 10 invasive alien plants, including *Solidago canadensis* L., *Avena fatua* L., and *Centaurea solstitialis* L., showed that the ensemble model performed excellently, with mean true skill statistics (TSS) and the area under the curve (AUC) both greater than 0.95 [35,36]. The results demonstrated the reliability of integrated models. Meanwhile, whether some environmental factors are considered during prediction significantly affects the results, mainly reflected in the ecological interpretation of species distribution. For instance, Xu [37] investigated the key environmental variables influencing *Aconitum leucostomum* Vorosch. and found that excluding environmental factors affecting species distributions during model building caused deviations in prediction results. In this study, the impacts of climate, terrain, soil, and human activities on the distribution of *X. spinosum* were comprehensively considered, and an ensemble model was used based on 5602 distribution data and 16 selected environmental factors. This study aimed to (1) predict and assess the potential geological distribution of *X. spinosum* under future climate scenarios, (2) determine the spread direction of *X. spinosum*, and (3) identify the dominant environmental factors affecting the distribution of *X. spinosum.* The results of this study not only provide scientific references for the monitoring, early warning, and control of *X. spinosum* in China but also offer methodological insights for the risk assessment of other invasive plant species. By predicting its potential distribution and spread directions, the findings can assist local governments and agricultural departments in identifying key areas for control, optimizing resource allocation, and mitigating the negative impacts of *X. spinosum* on both the ecological environment and the economy.

## 2. Materials and Methods

### 2.1. Data Collection and Processing for the Distribution of Xanthium spinosum L.

A total of 15,321 records, 10 records, and 422 records regarding geological distributions were retrieved from the Global Biodiversity Information Facility (GBIF, http://www.gbif.org/), the Chinese Virtual Herbarium (http://www.cvh.ac.cn), and the Southwest Environmental Information Network (http://swbiodiversity.org). Combined with 196 field survey records in China, 15,949 records of the global distribution of *X. spinosum* were obtained (Appendix A). The data from GBIF were processed using R 4.3.3 CoordinateCleaner [38]. To avoid spatial autocorrelation leading to model overfitting, the spThin package was used to retain only one distribution site within a 10 km range, and the obtained distribution sites were filtered. Finally, 5602 records were obtained for model construction (Appendix A).

### 2.2. Preprocessing of Environmental Data

The environmental variables included climatic factors, terrain factors, soil factors, and the human impact index (HII). In this study, current and future climatic factors, as well as terrain factors, were retrieved from the global climate database (http://www.worldclim.org/). The future climate data (2050s and 2070s) adopted four shared socioeconomic pathways (SSPs) (SSP1-26, SSP2-45, SSP3-70, and SSP5-85) under the Beijing Climate Center Climate System Model version 2 (BCC-CSM2-MR) for model prediction.

Soil factors (0–20 cm) were sourced from the Food and Agriculture Organization of the United Nations (https://www.fao.org/soils-portal/data-hub/soil-maps-and-databases/harmonized-world-soil-database-v20/en/, accessed on 10 June 2024). HII data were sourced from the Socioeconomic Data and Applications Center (SEDAC, https://sedac.ciesin.columbia.edu/data/set/wildareas-v2-human-influence-index-geographic/, accessed on 10 June 2024) of the National Aeronautics and Space Administration (NASA).

A total of 34 environmental (Appendix A) and climatic factors were obtained, and Arcgis 10.2 was used to standardize the resolution to 2.5 arcmin. Then, the obtained 34 environmental and climatic factors, along with the filtered distribution sites, were imported into R software (R 4.3.3). The Raster package was used to extract the corresponding climate data from these distribution sites. Pearson correlation coefficients of the variables were calculated to avoid excessively strong correlations among the environmental variables [39]. Meanwhile, considering the contribution of each variable, 16 environmental variables (Appendix A) with |r| < 0.7 were retained for subsequent modeling [40].

### 2.3. Model Prediction and Evaluation for the Suitable Habitats for Xanthium spinosum L.

The SDM integration platform biomod2, in the R environment, has been widely used for predicting species distribution [41,42,43,44]. Eight models built into biomod2 (ANN: Artificial Neural Network; CTA: Classification Tree Analysis; FDA: Flexible Discriminant Analysis; GAM: Generalized Additive Model; GLM: Generalized Linear Model; MARS: Multiple Adaptive Regression Splines; RF: Random Forest; and SRE: Surface Range Envelop or usually called BIOCLIM) were selected to model the suitable habitats for *X. spinosum*. First, biomod2 was used to generate three sets of pseudo-absence points, with each containing 1000 points [36,45,46]. For each single model, 75% of the data were randomly selected as the training dataset and 25% as the testing dataset; the process was repeated three times. TSS and AUC of the receiver operating characteristic (ROC) curve were used to evaluate the accuracy of the single models. TSS > 0.7 and ROC > 0.8 indicated good model performance [47,48]. The generated model results (e.g., RF, ANN, GLM, FDA, and MARS) used the weighted average method with TSS as the weight (EMwmByTSS) to construct an integrated model and then predict suitable habitats.

### 2.4. Distribution and Area of the Suitable Habitats for Xanthium spinosum L.

The suitable habitat prediction results generated using the integrated model under current and different future climate scenarios were imported into Arcgis 10.2 for normalization. A map of China was used for clipping. The natural breaks method was used to classify the species suitability levels into four categories: unsuitable habitats (0–0.3), low-suitability habitats (0.3–0.5), medium-suitability habitats (0.5–0.7), and high-suitability habitats (0.7–1.0) (Appendix A). Additionally, Raster reclassification was used to calculate the national territory area corresponding to each category.

### 2.5. Changes in the Spatial Patterns of the Suitable Habitats for Xanthium spinosum L.

Areas with species presence probability > 0.3 were classified as potential distribution areas. SDMtoolbox2.5 was used to classify the shift patterns of the suitable habitats for *X. spinosum* into expansion, unchanged, and contraction. The centroid position of *X. spinosum* in the future (2050s and 2070s) under different climate backgrounds was calculated, and the corresponding centroid shift trajectories were plotted.

## 3. Results

### 3.1. Evaluation of Model Accuracy

The evaluated performances of individual models are presented in Table 1. Among them, ANN, CTA, FDA, GLM, MARS, and RF demonstrated good performance and stability, each with a TSS > 0.75 and AUC > 0.9, indicating that SRE had the lowest TSS and AUC, indicating the worst performance. Additionally, GAM had AUC > 0.8 but TSS < 0.7, indicating slightly better performance than SRE. Additionally, the integrated model had a TSS of 0.826 and an AUC of 0.982, demonstrating high accuracy in predicting the potential species distribution.

### 3.2. Current Distribution of the Suitable Habitats for Xanthium spinosum L. in China

The suitable habitats for *X. spinosum* in China are widely distributed under current climatic conditions (Figure 1). Except for Heilongjiang and Jilin provinces, which have sporadic distributions, all other provinces and regions exhibit varying degrees of presence of *X. spinosum.* The high-suitability habitats are mainly distributed in southeastern Gansu, Shaanxi, southern Shanxi, most of Henan, Shandong, Jiangsu, northern Anhui, and Hubei. Additionally, some regions with higher altitudes, such as parts of Xinjiang, northern Yunnan, and eastern Guizhou, also have high-suitability habitats. The present study calculated that the current total area of suitable habitats was 414.7396 × 10^4^ km^2^, where the area of medium-suitability habitats was the largest (50.31%), amounting to 208.6410 × 10^4^ km^2^, and areas of high- and low-suitability habitats were 70.8056 × 10^4^ and 128.2930 × 10^4^ km^2^, respectively.

### 3.3. Future Distribution of Suitable Habitats for Xanthium spinosum L. in China

The area of suitable habitats for *X. spinosum* increased under four future climate scenarios (SSP1-26, SSP2-45, SSP3-70, and SSP5-85) compared with current climatic conditions (Figure 2 and Table 2). Specifically, the area increased, and the increment was maximized (67.9827 × 10^4^ km^2^) under SSP3-70 during the current–2070s. The area of high-suitability habitats decreased, and the decrement was maximized (51.2066 × 10^4^ km^2^) under SSP5-85 during the current–2070s. Meanwhile, the area of high-suitability habitats decreased under the four climate scenarios compared with current climatic conditions, except under the SSP2-45 scenario, where the area of medium-suitability habitats showed a slight increase, followed by a slight decrease over time. The area of low-suitability habitats gradually increased over time under the four future climate scenarios. Overall, the areas more suitable for *X. spinosum* growth shrank with climate and environmental changes. However, to avoid extinction, *X. spinosum* populations migrated to areas more conducive to their growth while adapting to the environment.

### 3.4. Future Distribution Patterns of the Suitable Habitats for Xanthium spinosum L. in China

The suitable habitats for *X. spinosum* expanded under the four future climate scenarios during the current–2050s. The expansion areas were mainly concentrated in Mongolia, Northeast China, and the plateau regions (e.g., Xinjiang and Xizang), whereas the contraction areas were concentrated in the southeast coastal area of China (Table 3 and Figure 3). During the 2050s–2070s, the suitable habitats for *X. spinosum* exhibited varying trends under different climate scenarios. Under SSP1-26, the suitable habitats in the previously expanding areas of Northeast China and Inner Mongolia began to contract, with contraction outpacing expansion. Under SSP2-45, these areas continued to expand, whereas the suitable habitats in Xinjiang and Gansu began to contract. Under SSP3-70, *X. spinosum* continuously expanded toward plateau regions including Northeast China, Inner Mongolia, Qinghai, Sichuan, and Xizang, whereas the suitable habitats in Guangxi, Guangdong, and Yunnan contracted. Finally, under SSP5-85, the suitable habitats in Jilin, Liaoning, Inner Mongolia, Hebei, and the southeast coastal area of China contracted, whereas the expansion mainly concentrated in the plateau regions (e.g., Qinghai and Xizang).

Under the current climatic conditions, the centroid of *X. spinosum* is located in Hanzhong, Shaanxi (106°59′50.72″ E, 33°10′52.93″ N). Projections based on future climate scenarios indicate that the centroid will shift either northwestward or northeastward (Figure 4). Under SSP1-26, the centroid of *X. spinosum* shifts northeastward by 96.439 km to Taibai County, Baoji City (107°41′28.91″ E, 33°49′38.73″ N) during the current–2050s period, then southwestward by 37.090 km to Liuba County, Hanzhong City (107°18′14.62″ E, 33°44′24.13″ N) during the 2050s–2070s. Under SSP2-45, it shifts by 161.990 km to Chencang District (106°51′8.29″ E, 34° 37′59.59″ N) during the current–2050s, then southeastward by 82.136 km to Qishan County, Baoji City (107°42′7.46″ E, 34°23′52.46″ N) during the 2050s–2070s. Under SSP3-70, the centroid of *X. spinosum* shifts northwestward by 75.088 km to Feng County, Baoji City (106°57′36.30″ E, 33°51′21.37″ N) during the current–2050s, then northeastward by 71.897 to Chencang District, Baoji City (107°24′41.52″ E, 34°23′0.86″ N) during the 2050s–2070s. Under SSP5-85, it shifts northeastward by 131.123 km to Weibin District, Baoji City (107°18′5.39″ E, 34°19′59.42″ N) during the current–2050s, then southwestward by 26.81 km (107°0′40.92″ E, 34°18′22.99″ N) during the 2050s–2070s.

### 3.5. Analysis of the Environmental Factors Influencing the Distribution of Xanthium spinosum L.

The results obtained using the biomod2 model, after performing importance ranking, showed that the five environmental factors with the greatest impact on the distribution of *X. spinosum* and their importance values were HII (0.04686), temperature annual range (bio7, 0.04685), maximum temperature of the warmest month (bio5, 0.03813), annual precipitation (bio12, 0.03329), and mean temperature of the driest quarter (bio9, 0.03108). In this study, HII, bio7, bio5, and bio12 collectively accounted for 61.3% of the total importance, suggesting that HII, bio7, bio5, and bio12 were the key environmental factors influencing the distribution of *X.* spinosum.

According to the response curves of the key environmental factors (Figure 5), the habitat suitability for *X. spinosum* gradually improved with an increase in HII and eventually stabilized at a certain level. The habitat suitability for *X. spinosum* showed a significant declining trend when the temperature annual range (bio7) was around 30 °C, the maximum temperature of the warmest month (bio5) was approximately 35 °C, and the annual precipitation (bio12) exceeded 700 mm. This suggested that, besides areas affected by human activities, *X. spinosum* was more suited to regions with an annual temperature below 30 °C, maximum temperature in the warmest month less than 35 °C, and annual precipitation between 0 and 700 mm.

## 4. Discussion

### 4.1. Assessment of Biomod2 Model Prediction Performance

The strengths and weaknesses of different modeling algorithms in species distribution modeling have been extensively discussed in the literature. However, no single model or class of models has emerged as the optimal solution capable of achieving the highest statistical accuracy across all species distribution studies [49]. While a single model may not always provide the best results in every case, the valuable information derived from individual models should still be taken into account. Ensemble models offer unique advantages in model selection methodologies, effectively mitigating issues related to ecological prediction variability inherent in various statistical approaches [50]. Furthermore, studies on species such as *Panax ginseng* C. A. Mey. and *Magnolia officinalis* Rehder & E. H. Wilson have shown that ensemble modeling approaches can improve model accuracy and reduce uncertainty to a certain extent [51,52]. This study was novel in using an integrated model to predict the potential geological distribution of *X. spinosum* in China and analyze its future distribution patterns and the key environmental variables affecting its distribution. The ensemble model (TSS: 0.826, AUC: 0.982) was compared favorably against eight individual models using the evaluation metrics TSS and AUC. As indicated, the TSS average of the model was slightly lower than its CTA (TSS: 0.8455) and RF (TSS: 0.9861); its AUC average was only slightly inferior to RF (AUC: 0.9999), However, the RF might have extreme values. Overall, the predictive accuracy of the model was superior to that of most single models, making it suitable for guiding the early monitoring and warning measures for *X. spinosum* in China.

### 4.2. Change Pattern and Shift Direction of the Suitable Habitats for Xanthium spinosum L. Under Future Climate Change

Biological invasion and climate change are considered two major driving factors for global biodiversity loss and ecosystem service changes [53,54,55]. Abnormal temperatures and precipitation resulting from climate change may affect ecosystem stability, thus offering more opportunities for invasion by alien species [56]. Hamit [57] et al. used MAXENT to predict the distribution and diffusion of *X. spinosum* in Xinjiang. They found that the area of suitable habitats for *X. spinosum* gradually expanded, which was consistent with the results of the present study. However, this study predicted a broader distribution area of *X. spinosum* in Xinjiang during different periods, which was possibly due to the inclusion of HII besides climate, terrain, and soil factors. Human activities are quite significant for invasive alien species. Shana et al. further explained the association between human activities and the Allee effect in the emerald ash borer (*Agrilus planipennis* Fairmaire) populations, confirming the importance of human activities in the reproductive success of this species [58,59,60]. Another study on *Poa annua* L. found that the inhabited Marion Island with *P. annua* had higher genetic diversity and gene flow with congeneric plants on the island compared with the uninhabited Prince Edward Island [61]. The area of high-suitability habitats of *X. spinosum* continuously decreased over time under future climate scenarios compared with the current climate scenario. However, its distribution also became more dispersed and fragmented, especially under the SSP5-8.5 scenario. This might be attributed to drastic climate changes under global warming, resulting in reduced species diversity, exacerbated ecosystem fragmentation, and increased extinction risks for endangered species [62,63]. Meanwhile, climate change creates favorable conditions for invasive alien plants to expand their range into higher latitudes [64], as demonstrated by the centroid shift direction of *X. spinosum*. Similar migratory trends have also been noted in species distribution studies of *Luculia pinceana* Hook and *Phyllostachys edulis* (Carrière) J. Houz. [65,66].

Additionally, the expansion areas of *X. spinosum* are concentrated in high-altitude regions such as Qinghai and Xinjiang. This may be because climate change alters the mountain ecosystem and intensifies invasion pressure [67]. Under the framework of niche conservatism, invasive species often retain the fundamental characteristics of their native ecological niches in new environments. This trait enables them to rapidly adapt to ecosystem shifts driven by climate change [68]. In alpine regions, the warming climate increases the availability of suitable niches, offering more opportunities for species expansion. As observed from the centroid shift center, the centroid of *X. spinosum* generally shifts northward, though varying degrees of reverse shifts occur under all climate scenarios except SSP3-7.0. The regions experiencing shifts are located within Baoji City, which is surrounded by mountains on three sides and is a typical fold mountain region [69]. Generally, invasive alien plants shifting from low-altitude regions to mountainous regions are restricted by low temperatures [70], and microtopography (slope and aspect) can provide footholds or shelters for non-native species under above-ideal climatic conditions [71]. However, the impact of geographical barriers cannot be eliminated, which still restricts the shift and distribution of *X. spinosum*, thus explaining the backward shift of the centroid.

### 4.3. Human Activities and Climatic Factors Influencing the Distribution of Xanthium spinosum L.

Among various environmental factors, HII, bio7 (temperature annual range), bio5 (maximum temperature of the warmest month), and bio12 (annual precipitation) were the key environmental factors influencing the distribution of *X. spinosum*. The significant role of temperature in influencing species distribution was evident [72]. However, species invasion is a complex process that involves not only environmental factors such as temperature but also other key drivers. Among these, propagule pressure and introduction pathways play particularly important roles in determining the success of invasion [73]. High propagule pressure, characterized by the input of a large quantity of seeds or reproductive materials, can significantly increase the probability of species establishment in new regions. For some invasive species, a single introduction with sufficient quantity can establish a self-sustaining population [41]. Moreover, diverse introduction pathways can further impact the dispersal distances of species. These pathways include natural dispersal (e.g., wind, water) and human activities (e.g., agricultural transport, traffic). It is noteworthy that human activities mediated transportation, migration, and commercial activities promote introductions into new remote areas and accelerate the spatial expansion of species after their introduction into new areas [74,75]. Therefore, the prickly seeds of *X. spinosum* are easy to carry around by human activities and spread far away. In addition, according to the response curves of bio5 and bio7, especially bio5, the habitat suitability of the species displayed an inverted U-shape, and the response ranged from −10 °C to 40 °C, suggesting that *X. spinosum* could tolerate a wide temperature range, which was consistent with its biological characteristics of good cold resistance and heat tolerance [13]. Precipitation can control the aboveground biomass of plants by affecting nitrogen deposition levels and is an essential factor influencing plant community growth and composition [76]. Soil and terrain factors were the least important among the environmental factors selected in this study. In contrast, climatic factors such as temperature and precipitation contributed to the distribution patterns of *X. spinosum* under different climate scenarios [37]. Human activities further promote the migration and diffusion of *X. spinosum*.

*X. spinosum* exhibits strong adaptability and reproductive capacity, posing significant challenges for management once it successfully invades a new area. The plant’s tall stature and thorny characteristics hinder mechanical removal, while manual removal, although thorough, is time-consuming, labor-intensive, and costly [12]. Moreover, the lack of targeted chemical control agents for cocklebur further complicates management efforts [77]. Therefore, it is critical to prioritize monitoring and early-warning systems based on habitat suitability predictions when addressing *X. spinosum* invasions. Habitat suitability modeling can identify high-risk regions, such as Inner Mongolia, Northeast China, and the Qinghai–Tibet Plateau. By establishing monitoring networks in these areas in advance, the goals of “early detection and early management” can be achieved, effectively mitigating the ecological and economic risks posed by *X. spinosum* invasion. Rather than incurring substantial post-invasion management costs, it is more effective to prevent invasions at the source through ecological risk assessments and control of dispersal pathways. Against the backdrop of global climate warming, the potential distribution range of *X. spinosum* may further expand, underscoring the need for strengthened monitoring and preventive measures to mitigate risks proactively.

## 5. Conclusions

This study constructed an integrated model based on eight single models in biomod2, with results demonstrating strong predictive performance. The predictions indicate that under current climate conditions, the potential geographical distribution of *X. spinosum* in China spans an area of 414.7396 × 10⁴ km^2^, primarily concentrated in regions south of the Yellow River and the eastern parts of Sichuan and Yunnan provinces. In contrast to current climate conditions, the total area of suitable habitats for *X. spinosum* is projected to increase in the future. From a spatial pattern perspective, the suitable habitats for *X. spinosum* are expected to expand towards Inner Mongolia, the northeastern provinces, and high-altitude areas such as Xinjiang and Tibet. Regarding centroid migration, suitable habitats for *X. spinosum* are gradually shifting towards higher latitudes, with the largest migration occurring under the SSP2-45-2050s scenario, covering a distance of 161.990 km. Key environmental factors influencing *X. spinosum* distribution include temperature (bio5, bio7), precipitation (bio12), and human activity (HII). This study provides critical theoretical and practical insights for monitoring, early warning, and the scientific management of *X. spinosum* expansion.

## Figures and Tables

**Figure 1 plants-14-00306-f001:**
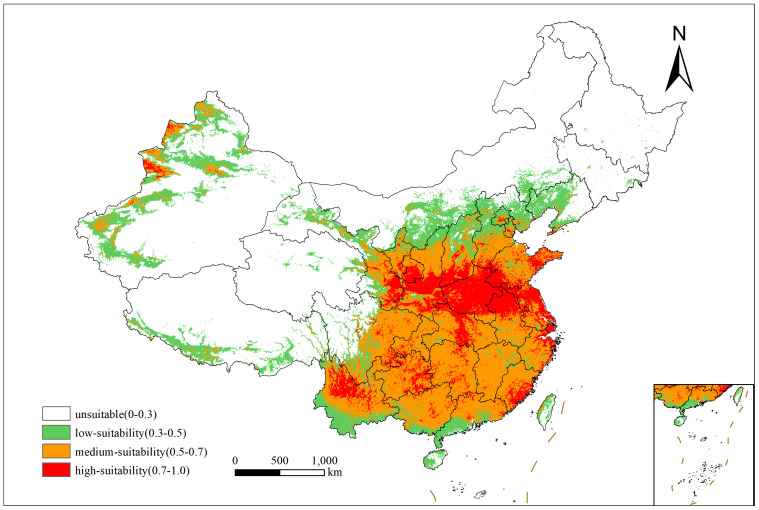
Current suitable distribution area of *Xanthium spinosum* in China. White indicates an unsuitable growth area, and red indicates a high-suitability area.

**Figure 2 plants-14-00306-f002:**
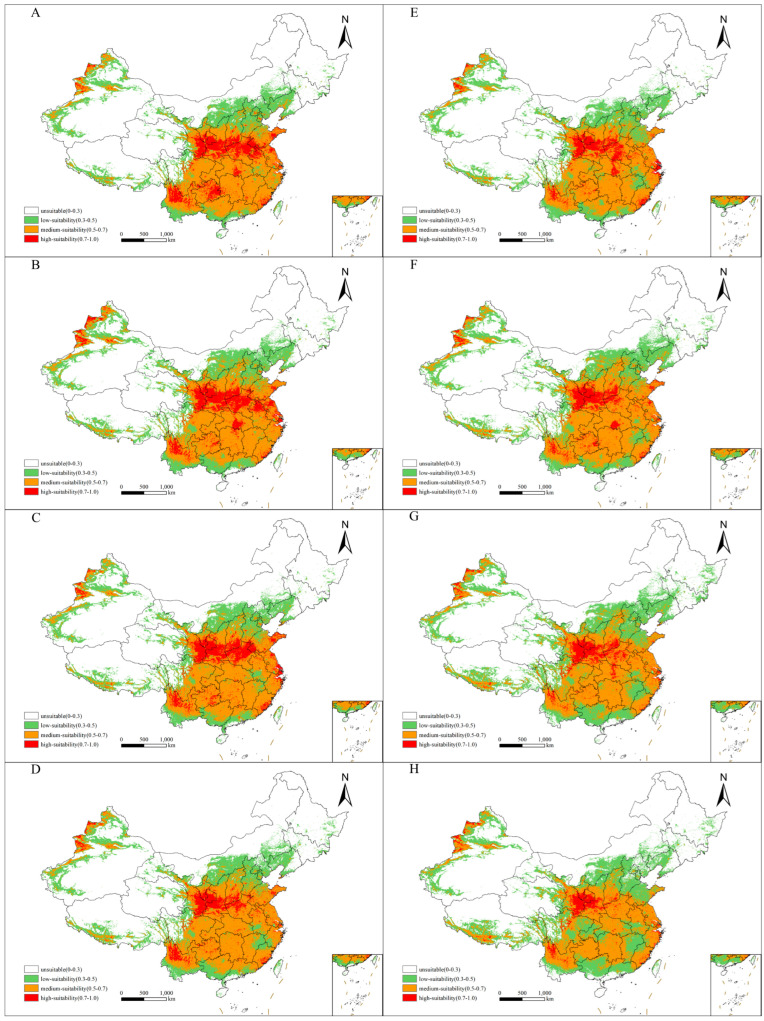
Potential distribution of *Xanthium spinosum* during different periods in China. (**A**–**D**): SSP1-26, SSP2-45, SSP3-70, SSP5-85 in 2050s; (**E**–**H**): SSP1-26, SSP2-45, SSP3-70, SSP5-85 in 2070s.

**Figure 3 plants-14-00306-f003:**
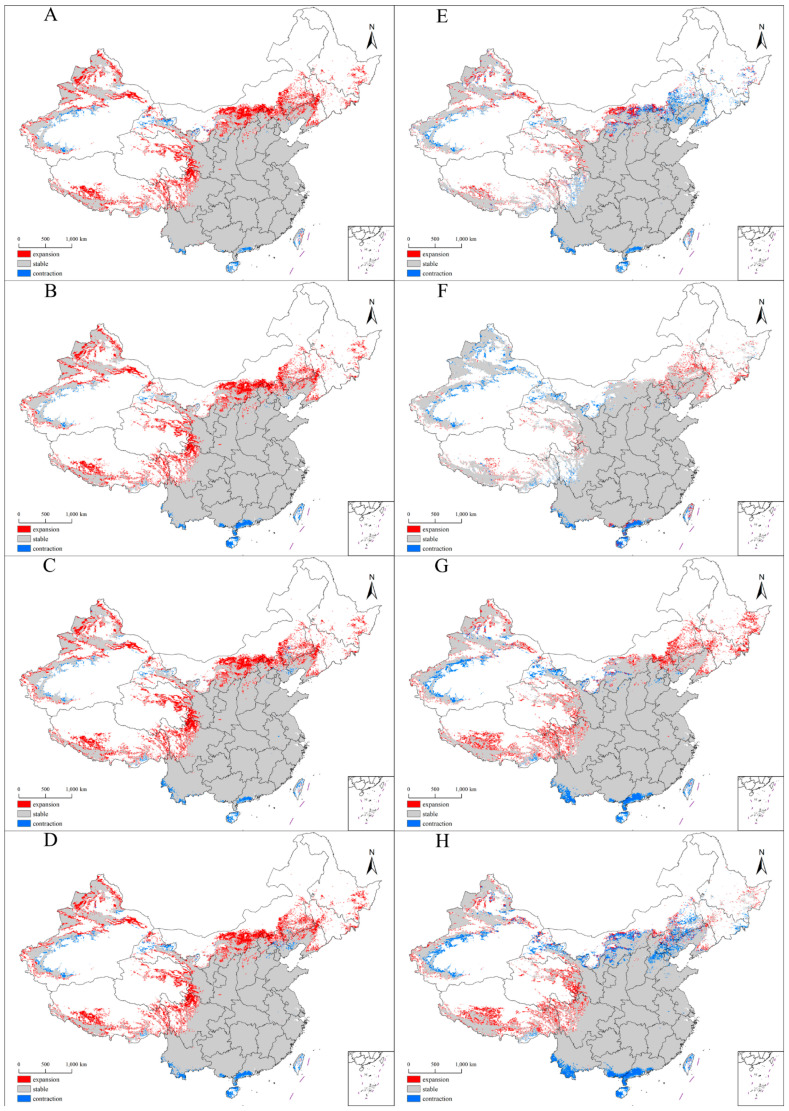
Trends of the spatial distribution pattern of *Xanthium spinosum* under different climate scenarios. (**A**–**D**): current vs. 2050s, SSP1-26, SSP2-45, SSP3-70, SSP5-85; (**E**–**H**): 2050s vs. 2070s, SSP1-26, SSP2-45, SSP3-70, SSP5-85.

**Figure 4 plants-14-00306-f004:**
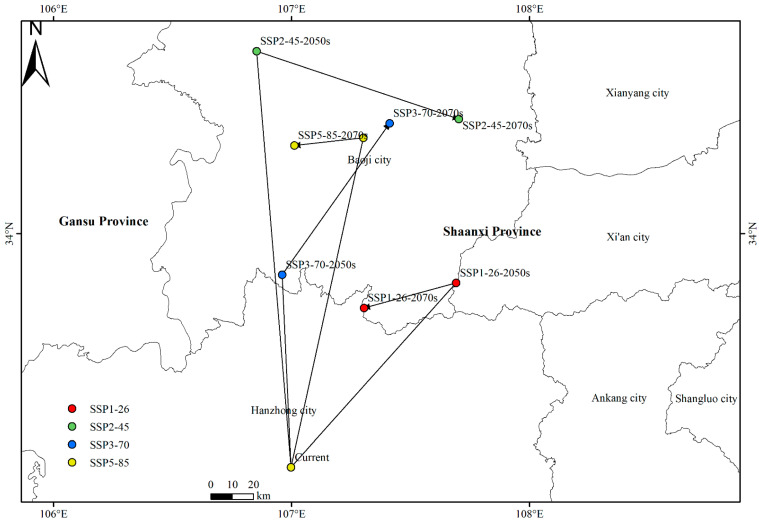
Centroid shifts under different climate scenarios for *Xanthium spinosum*.

**Figure 5 plants-14-00306-f005:**
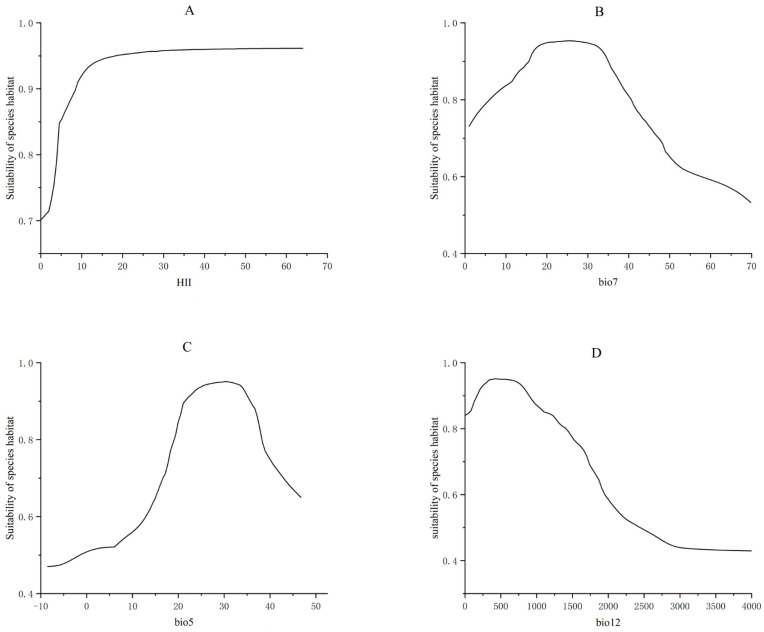
Response curves for the main environmental variables. (**A**) Human influence index; (**B**) Temperature annual range; (**C**) Maximum temperature of the warmest month; (**D**) Annual precipitation.

**Table 1 plants-14-00306-t001:** Accuracy evaluation results of different models (SD: Standard deviation; CV: Coefficient of variation.).

Models	TSS			AUC		
	Mean	SD	CV	Mean	SD	CV
ANN	0.7798	0.0132	0.0170	0.9471	0.0057	0.0060
CTA	0.8455	0.0301	0.0356	0.9528	0.0105	0.0110
FDA	0.7812	0.0079	0.0102	0.9516	0.0035	0.0037
GAM	0.6821	0.0092	0.0135	0.9200	0.0039	0.0043
GLM	0.7858	0.0075	0.0096	0.9571	0.0016	0.0017
MARS	0.7930	0.0070	0.0089	0.9574	0.0023	0.0024
RF	0.9861	0.0044	0.0044	0.9999	0.0003	0.0003
SRE	0.5680	0.0052	0.0092	0.7841	0.0028	0.0035

**Table 2 plants-14-00306-t002:** The areas of potential suitable habitats for *Xanthium spinosum* under different climate scenarios (×10^4^ km^2^).

Climate Scenario	Period	High-Suitability Habitat	Medium-Suitability Habitat	Low-Suitability Habitat	Total Suitable Habitat
Current	1970–2000	77.8056	208.6410	128.2930	414.7396
SSP-1-26	2050s	60.3160	227.5660	161.5590	449.4410
	2070s	41.0694	220.1440	174.2380	435.4514
SSP-2-45	2050s	61.2865	230.8140	174.4440	466.5445
	2070s	42.5226	243.8400	185.0760	471.4386
SSP-3-70	2050s	53.9792	231.4700	171.8660	457.3152
	2070s	37.2483	226.3350	219.1390	482.7223
SSP-5-85	2050s	40.1059	238.7590	179.2450	458.1099
	2070s	26.5990	207.4980	224.1910	458.2880

**Table 3 plants-14-00306-t003:** Spatial changes in the suitable areas for *Xanthium spinosum* in different climate scenarios.

Climate Scenario		Area (×10^4^ km^2^)			Change (%)		
		Expansion	Unchanged	Contraction	Expansion	Unchanged	Contraction
Current vs. 2050s	SSP1-26	41.24	408.20	6.54	9.94	98.42	1.58
	SSP2-45	59.93	406.62	8.12	14.45	98.04	1.96
	SSP3-70	50.12	407.20	7.54	12.08	98.18	1.82
	SSP5-85	54.12	403.99	10.75	13.05	97.41	2.59
2050s vs. 2070s	SSP1-26	2.43	433.02	16.42	0.54	96.35	3.65
	SSP2-45	13.05	458.39	8.15	2.80	98.25	1.75
	SSP3-70	39.81	442.91	14.41	8.71	96.85	3.15
	SSP5-85	29.19	429.10	29.01	6.37	93.67	6.33

## Data Availability

Data are contained within the article and Appendix A.

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
