# Peer review of "Impacts of Human Activity and Climate Change on the Suitable Habitats for Xanthium spinosum in China"

_plants, 2025, doi:10.3390/plants14030306_

Round 1
Reviewer 1 Report
Comments and Suggestions for Authors Manuscript Liu et al. relates to a current area of ​​research -biogeography and invasion of one of the species of terrestrial plants,
Xanthium spinosum. The authors link the spread of this species to
anthropogenic impacts and climate warming and, based on the model,
predict its wider distribution by 2050 and by 2070. Previously published
invasion models (Yuan et al., 1917; Tao et al., 2020; Xiao et al., 2023)
did not consider the potential geological distribution of this species in
response to climate change. The merit of the authors is that they close
this gap with their work, and their results can be used for monitoring
purposes. At the end of the introduction, the hypothesis is not formulated,
but three goals are stated. They have been achieved, and the Conclusion is
based on the results obtained by the authors. Of the 76 literature references,
30% refer to the years 2020-2024. Self-citations are about 12%. The manuscript
is well structured, containing maps and tables from which the essence of the work
is visible. There are no comments on their design. In addition to the maps, at the
end of the article a clear diagram of trends under different scenarios is given. The
methods are well described, with references to literature sources. There are some
errors in the text with the citation of literary sources and the references are not
carefully formatted.
Author Response
Comments:Manuscript Liu et al. relates to a current area of ​​research -biogeography and invasion of one of the species of terrestrial plants,Xanthium spinosum. The authors link the spread of this species to anthropogenic impacts and climate warming and, based on the model,predict its wider distribution by 2050 and by 2070. Previously published invasion models (Yuan et al., 1917; Tao et al., 2020; Xiao et al., 2023) did not consider the potential geological distribution of this species in response to climate change. The merit of the authors is that they close this gap with their work, and their results can be used for monitoring purposes. At the end of the introduction, the hypothesis is not formulated,but three goals are stated. They have been achieved, and the Conclusion is based on the results obtained by the authors. Of the 76 literature references,30% refer to the years 2020-2024. Self-citations are about 12%. The manuscript is well structured, containing maps and tables from which the essence of the work is visible. There are no comments on their design. In addition to the maps, at the end of the article a clear diagram of trends under different scenarios is given. The methods are well described, with references to literature sources. There are some errors in the text with the citation of literary sources and the references are not carefully formatted.
Response:
Thank you for your valuable suggestions regarding the references in our manuscript. We have carefully reviewed all the references used and made the following improvements:1)Corrected inconsistencies in the formatting of author names and ensured uniformity throughout the references.2)Added DOI numbers for several references to improve accessibility and traceability.3)Revised citations for books to align with the appropriate referencing standards.
If you notice any further issues or areas for improvement in the submitted manuscript, please do not hesitate to contact us. We are committed to making additional revisions as needed to ensure the highest quality of our work.
Reviewer 2 Report
Comments and Suggestions for Authors
Here are some suggestions for the authors:
The introduction is quite short and has to be revised
Table titles should be more detailed, providing information needed for someone to understand what the table presents.
Line 260: correct bibliography
Line 352: what are the 34 environmental and climatic factors
Conclusion: I suggest rewriting the conclusion in order to present them more clearly and in detail.
References
Some references are not appropriately cited from what I could assess, so this section should be thoroughly and carefully double-checked and revised to follow the mdpi reference style.
Comments on the Quality of English LanguageMinor editing of English language required
Author Response
Comments1:The introduction is quite short and has to be revised
Response1:We appreciate your suggestion to expand the Introduction. Additional content has been added in the following sections: line 72–78, line 95–101, and line 120–126 of the revised manuscript.
Comments2:Table titles should be more detailed, providing information needed for someone to understand what the table presents.
Response2:To improve clarity, we have revised the titles of several tables to provide more detailed information, enabling readers to better understand their content.line203,line245
Comments3:Line 260: correct bibliography
Response3:The bibliography issue in line 260 has been corrected. You can review the updated reference formatting in lines 607–608 of the revised manuscript.
Comments4:Line 352: what are the 34 environmental and climatic factors
Response4:Thank you for pointing this out. We have included detailed information about the 34 environmental and climatic factors in Supplementary Table 2, which is now available for your review.
Comments5:Conclusion: rewrite for clarity and detail
Response5:Based on your suggestion, we have revised the Conclusion section to present the findings more clearly and in greater detail. Please refer to lines 418–432 in the revised manuscript for the updated content.
Comments6:References: revise for consistency and MDPI style
Response6:We have thoroughly reviewed the references, addressing issues such as inconsistencies in author name formatting and ensuring adherence to the MDPI reference style. Additionally, DOI numbers have been added where applicable, and citations for books have been revised for accuracy.
If you find any additional issues or areas for improvement in the revised manuscript, please do not hesitate to contact us. Your feedback is invaluable in ensuring the quality and rigor of our work.
Thank you once again for your time and effort in reviewing our manuscript.
Reviewer 3 Report
Comments and Suggestions for Authors
It's a good and interesting paper.
Section 4. Materials and methods. In my opinion it should be before the results section. It would make it easier to know how the results have been obtained.
The author of the species is only indicated the first time it is written in the text. Likewise the genus (Xanthium) is only written the first time, then it can be written as X. on all occasions, always in italics.
It's necessary to add the author of the other species indicated in the text (lines 67, 68, 69, 92, 98, 227, 264).
Author Response
Comments1:Section 4. Materials and methods. In my opinion it should be before the results section. It would make it easier to know how the results have been obtained.
response1:We completely agree with your suggestion to place the "Materials and Methods" section before the "Results" section. This change has been made in the revised manuscript to improve readability and facilitate understanding of how the results were obtained.
Comments2:
The author of the species is only indicated the first time it is written in the text. Likewise the genus (Xanthium) is only written the first time, then it can be written as X. on all occasions, always in italics. It's necessary to add the author of the other species indicated in the text (lines 67, 68, 69, 92, 98, 227, 264).
response2:We have standardized the formatting of species names throughout the manuscript. The full species name, including the author, is provided the first time it is mentioned. For subsequent mentions, the genus is abbreviated as X. and is consistently italicized. Additionally, we have added the author information for other species mentioned in the text. You can review these changes at the following locations: line 69–70, line 106, line 112, line 316, and lines 353–354 in the revised manuscript.